# B-Mode and Doppler Ultrasonographic Findings of Prostate Gland and Testes in Dogs Receiving Deslorelin Acetate or Osaterone Acetate

**DOI:** 10.3390/ani10122379

**Published:** 2020-12-11

**Authors:** Wojciech Niżański, Małgorzata Ochota, Christelle Fontaine, Joanna Pasikowska

**Affiliations:** 1Wrocław University of Environmental and Life Sciences, Department of Reproduction and Clinic of Farm Animals, pl. Grunwaldzki 49, 50-366 Wrocław, Poland; malgorzata.ochota@upwr.edu.pl; 2Virbac Group, Global Marketing and Business Optimization Department, Companion Animals Veterinary Exclusive Ranges Section, 13ème rue LID, 06511 Carros, France; christelle.speiser-fontaine@virbac.com; 3Veterinary Clinic, B. Krzywoustego Str. 105/22, 51-166 Wroclaw, Poland; j.h.pasikowska@gmail.com

**Keywords:** dog, prostate, benign prostatic hyperplasia, ultrasound

## Abstract

**Simple Summary:**

This article presents the use of ultrasonography (basic and advanced techniques) in dogs suffering from benign prostate hyperplasia (BPH) for the assessment of the effects of two different medications: osaterone acetate (Ypozane^TM^), a registered drug for BPH in dogs, and deslorelin acetate (Suprelorin^TM^), registered for pharmacological castration in dogs. Based on the obtained results it can be stated that both B-mode and color Doppler Ultrasound imaging techniques are suitable for diagnosis and progress assessment of dogs suffering from BPH. Both investigated medications (osaterone acetate and deslorelin acetate) led to a significant sonographic improvement. Deslorelin acetate reduced prostate volume more slowly, but its effect lasted longer than for osaterone acetate.

**Abstract:**

This article presents B-mode and color Doppler imaging of the prostate and testes in dogs suffering from benign prostate hyperplasia (BPH), and receiving deslorelin acetate (Suprelorin^TM^) or osaterone acetate (Ypozane^TM^). The study was planned as a controlled clinical trial, dogs were divided into negative control (healthy dogs, *n* = 10), positive control (dogs with BPH, *n* = 10), and study groups, III (*n* = 15), receiving deslorelin acetate (DA), and IV (*n* = 10), receiving osaterone acetate (OA). The B-mode appearance of the prostate parenchyma improved in all investigated dogs from the DA group, and in 60% of OA dogs. Prostate volume was reduced more quickly with OA (from D14), but lasting for a shorter time (on average up to week 20), compared to DA that reduced the prostate volume more slowly (>8 weeks), but the reduction remained longer (>24 weeks). The systolic peak velocity (SPV) and mean velocity (Vmean) were higher in all dogs diagnosed with BPH, compared to Control Group I. The indices did not change in both Control Groups I and II, whereas in study Groups III and IV they decreased throughout the study period compared to day 0 and Control Group II. In Group III the highest reduction was noted from day 21 to week 8, whereas in Group IV the lowest Vmean was recorded before day 21. Testicular parenchyma and volume changed significantly in Group III receiving DA, and the velocity of blood flow in the testicular artery correlated positively with testicular volume only in this group (III). The present study proved the usefulness of B-mode and color Doppler US imaging techniques for diagnosis and progress assessment of dogs suffering from BPH. The blood flow kinetics (mainly SPV) demonstrated a time association between the blood flow changes registered in the prostatic artery, and the subsequent volumetric and sonographic improvement of the prostate parenchyma. The reduction in flow indices was noted prior to the reduction in prostate volume, suggesting that the sonographic recovery of the prostate tissue, occurs secondarily to the regression of the prostate vascular system. Both investigated medications (osaterone acetate and deslorelin acetate) led to a significant sonographic improvement. Deslorelin acetate reduced prostate volume more slowly, but its effect lasted longer than for osaterone acetate.

## 1. Introduction

Imaging is a very accurate tool for the identification of disease in patients suspected of reproductive tract disorders. B-mode ultrasonography allows for a detailed assessment of the inner structure, whereas Doppler ultrasonography accurately characterizes the nature of blood flow and organ vasculature. In dogs, the prostate is a bilobed gland, located astride the prostatic urethra and surrounded by a capsule [1]. Its size varies significantly between individuals, and often increases with age in non-castrated dogs. The first clinical manifestation of prostatic disease, apart from the gland enlargement, are the changes to echogenicity and heterogeneity of the gland parenchyma. Ultrasonography of the testes allows for the detection of palpable and non-palpable changes, or differentiating testicular from epididymal and scrotal diseases [1]. Although for a definitive diagnosis of prostate pathology transabdominal prostatic biopsy is the gold standard, a typical ultrasonographic picture, together with the clinical findings, are usually enough for a presumptive diagnosis, and in some cases suitable treatment [2]. More recently, Doppler ultrasonography has been utilized for providing useful data on blood flow patterns and velocity in relation to prostate gland status, or the association between testicular flow patterns and semen quality [3,4]. In human medicine color Doppler ultrasound is used to assess blood flow in prostatic and testicular arteries, helping in diagnosing prostate and testicular pathologies, and in predicting the level of spermatogenesis [5]. Doppler evaluation of dog reproductive organs has recently been performed, providing useful information about blood flow and velocity, and their relation to semen quality [3,6].

In dogs, one of the most common prostate diseases is benign prostate hyperplasia (BPH). It is a testosterone dependent, non-cancerous prostate enlargement. Androgenic stimulation causes the prostatic cells to proliferate (hyperplasia) and increase in size (hypertrophy). The condition may remain unnoticed for long periods of time before the marked enlargement of the gland leads to the symptomatic stage. The severity of clinical signs are individually dependent, and dogs are usually presented with sero- to sanguineous preputial discharge independent of urination, hematuria, ribbon-like stools, and constipation [7,8]. In dogs with BPH, pharmacological therapy should be considered, especially in valuable studs, where preserving fertility is crucial. Besides human medications such as 5-alfa-reductase inhibitors (finasteride), androgen receptor blockers (flutamide), or hormones (progestogens, estrogens and antiestrogens), the only medication registered for BPH treatment in dogs is osaterone acetate (OA, Ypozane^TM^, Virbac) [9]. It contains a steroidal androgen which treats BPH by reducing the uptake of androgens in the prostatic tissue and inhibiting the action of 5a-reductase [10]. However, another drug has recently been proposed to alleviate BPH symptoms in dogs; the GnRH agonist, deslorelin acetate (DA, Suprelorin^TM^, Virbac). It is a GnRH superagonist, which after the initial flare-up leads to desensitization of the pituitary receptors responsible for gonadothropin release, and a significant decline in LH and FSH, which in turn reduces androgens production [11]. The clinical outcome is very similar to the effect seen with surgical castration. DA markedly reduces the prostate size, but it also affects libido and sperm production. Both medications effectively alleviate clinical symptoms related to prostate enlargement, however the available literature provides only limited data concerning the ultrasonographic changes in the prostate and testes of dogs with BPH and receiving both drugs [12,13,14,15].

Although OA and DA have been used in dogs for some time now, there has been no publications on how these drugs affect the echogenicity and blood flow during long-term use. Both medications have a completely different mode of action, OA being antiandrogen and not interfering with fertility, whereas DA contributes to the quiescence of the hypothalamic–pituitary–gonadal axis. Ultrasonographic monitoring would provide detailed information on the onset and length of action, efficacy, and potential drawbacks of each medication. Thus, the objective of the current study was to analyze the B-mode and color Doppler changes of the prostate gland and testicles in healthy dogs, dogs with BPH, and dogs with BPH receiving osaterone acetate or deslorelin acetate. The study was planned as a controlled clinical trial, under the good clinical practice (GCP) guidelines, and aimed to compare the ultrasound findings in OA or DA use, based on an evidence based medicine approach.

## 2. Materials and Methods

The trial was approved by the II Local Ethics Committee for Animal Experiments of the University of Environmental and Life Sciences in Wroclaw (No 36/2014).

### 2.1. Animals Selection

A total of 45 intact male dogs, aged more than 5 years and of different breeds, were enrolled for the study. Dogs were divided into four groups (two control and two study groups).

*Control groups:* Group I (negative control group) comprised 10 healthy dogs aged 5–10 years (mean 6.5; SD ± 2) and weighing 8.7–23.5 kg (mean 17.08; SD ± 6.21). Group II (positive control group) included 10 dogs, aged 5–15 years (mean 9.5; SD ± 3.5) and weighing 2.9–44 kg (mean 14.12; SD ± 12.17), diagnosed with BPH, and for which owners decided against treatment due to different reasons (valuable semen donors, working, racing or hunting dogs).

*Treated groups:* Group III (deslorelin acetate group) included 15 dogs aged 6–15 years (mean 9.47; SD ± 2.13), weighing 7.6 and 46.5 kg (mean 27.6; SD ± 14.65), diagnosed with BPH and treated with deslorelin acetate. Group IV (osaterone acetate group) comprised 10 dogs, aged 5–10 years (mean 7.6; SD ± 1.96) and weighing 4.1–50.5 kg (mean 26.92; SD ± 15.25), diagnosed with BPH and treated with osaterone acetate.

### 2.2. Ultrasonographic Study

All scanning procedures were performed by the same person to minimize operator-related variance in the results. The investigated dogs were examined without sedation, positioned in lateral recumbency on a table with a mat underneath. A MyLab25 Gold (Esaote, Italy) ultrasound system equipped with 12 MHz-micro convex probe and 12 MHz- linear probe was used. For the prostate imaging the transabdominal, prepubic approach was applied with the transducer placed on the caudoventral abdomen close to the prepuce, whereas for the testicles a straightforward prescrotal procedure was applied.

### 2.3. B-Mode Ultrasonography—Prostate Gland

During the first examination (Day 0, D0) the whole image of the prostate with margins and symmetry, as well as medial iliac lymph nodes and the presence of the periprostatic cysts were assessed. During each scanning four thematic categories, concerning B-mode prostate parenchyma appearance, were identified, and its particular aspects were described using exact, descriptive terminology [14]: (1) background echo texture (normal, hyperechoic, hypoechoic); (2) parenchymal stippling (regular, increased, coarse); (3) general appearance (homo- and heterogenous); and (4) focal changes (cysts, mineralized opacities, focal hypoechoic lesions). To facilitate the evaluation of the results, all data found during each examination was tabulated (Table 1). Moreover, during each examination the length (L), depth (D), and width (W) of the prostate gland were measured, and the prostate volume (VP) was calculated using the formula published by Atalan et al. [16]:VP (cm^3^) = 0.487 × L × W × (DL + DP): 2 + 6.38(1)

### 2.4. Doppler Ultrasonography—Prostate Gland

The blood flow of the prostatic artery was imagined using color Doppler (CD) and pulse wave Doppler (PW) sonography, and the blood flow patterns were recorded by pulse wave Doppler (PW) sonography. The prostatic artery was approached from the lateral location, before entering the gland, as suggested by Guenzel-Apel et al. [12]. The scanning window and the pulse repetition frequency (PRF) were adjusted to the lowest possible values without aliasing artifacts, and the Doppler gain was adjusted accordingly, so as not to show the background color noise. The minimum sample gate setting for this unit (2–4 mm) was used to record the blood flow pattern, and the angle of insonation was <45°. Three good quality measurements were analyzed for mean values of systolic peak velocity—SPV, mean velocity—Vmean, and resistive index—RI.

### 2.5. B-Mode Ultrasonography—Testes

The whole testes were scanned starting from the left testicle, in the sagittal and transverse planes. During each scanning procedure the following features concerning B-mode testicular parenchyma were evaluated [1]: background echotexture (classified as normal, hyper- or hypoechoic), mediastinum appearance (normal, widened, or atrophied), general appearance (hetero- or homogenous), and the presence of focal changes (focal hypo and hyperechoic lesions, cyst). The mean testicular volume of both testicles was calculated using the following formula [17] during each scan:VT (cm^3^) = 0.5 L × 0.5 W × 0.5 D × 4/3 π(2)

### 2.6. Doppler Ultrasonography—Testes

The blood flow in the testicular artery was imaged using color Doppler (CD) and pulse wave Doppler (PW) sonography. The testicular artery was imaged supratesticulary, between the testes and epididymis in the caudal testicular pole [12]. The scanning window and the pulse repetition frequency (PRF) were adjusted to the lowest possible values without aliasing artifacts, and the Doppler gain was adjusted accordingly, so as not to show background color noise. The minimum sample gate setting for this unit (2 mm) was used to record blood flow pattern, and the angle of insonation was <45°. Three good quality measurements were analyzed for mean values of systolic peak velocity—SPV, mean velocity—Vmean, and resistive index—RI.

### 2.7. Indications for Inclusion and BPH Confirmation

Dogs were allocated into the Control Group II, and Study Groups III and IV based on clinical criteria presented in Niżański et al [18], the initial B-mode appearance of the prostate gland [14], and US-guided FNA (fine needle aspirate) of the prostate in order to confirm BPH by cytological criteria [19].

### 2.8. Drugs and Schedule

Group III on D0, after performing the FNA of the prostate, received a 4.7 mg deslorelin acetate implant (Suprelorin^TM^, Virbac, Carros, France), inserted subcutaneously on the dogs’ back.

Group IV on D0, after performing the FNA of the prostate, were started on the osaterone acetate (Ypozane^TM^, Virbac, Carros, France) tablets, administered orally at a dose recommended by the manufacturer: 0.25–0.5 mg/kg every 24 h for 7 consecutive days.

### 2.9. Study Design

The trial was performed as a clinical, monocentric, randomized, and not-blinded study. The patients were allocated into one of the II–IV groups and received OA or DA as described above (Group III and IV), or were not treated and served as positive control group (Group II).

All control and study dogs were scanned on the following days: Day 0 (D0); Day 7 (D7); Day 14 (D14); Day 21 (D21); and Week + 8 (W8); Week + 12 (W12); Week + 16 (W16); Week + 20 (W20); and in Week + 24 (W24). Dogs from Groups III and IV underwent the follow-up examination 3 months later, i.e., after the manufacturer’s guaranteed duration of action for both drugs, to evaluate potential long-term effects of both drugs (Week + 36, W36). Each time the prostate gland and testes were evaluated using B-mode and Doppler imaging.

The obtained results were compared with respect to the D0 values in individual groups, then to values obtained in the Control Group II (positive control group), and finally the results were also compared between Group III and IV to see differences between both investigated drugs.

For comparison of prostate volume between D0 and the following measurements obtained in individual dogs, the absolute values were used. However, to reliably monitor the differences between the initial and the following measurements in the study groups, due to breed and size differences, the absolute values were converted into percentages, assuming D0 value as 100%. The changes noted during the study were analyzed in relation to this initial value, estimated as 100%.

### 2.10. Statistical Analysis

Statistical analysis was carried out using the STATISTICA 10.0 software (StatSoft, Kraków, Poland). Descriptive statistics, such as the mean and SDs, were calculated for each prostate measurement. The Shapiro–Wilk test was carried out to verify whether the parameters had a normal distribution. The values within the study groups with respect to D0 and the values between control and study groups were compared using the paired Student’s t-test, or U Mann–Whitney test. The natural anatomical differences in prostatic and testicular sizes in individual dogs were addressed using percentage (%) instead of numeric values (cm^3^), and the D0 volume was considered 100% [20]. The relationship between prostate size and blood flow in the prostatic and testicular arteries in Group III and IV were assessed using Pearson’s coefficient test. A *p* = 0.05 probability value was considered statistically significant.

## 3. Results

### 3.1. Prostate B-Mode Appearance

The prostate parenchyma in healthy dogs (Group I) was moderately and uniformly normoechoic, with medium to fine texture. The urethra was imaged in the center of the prostate, as a hypo- to anechoic structure, with a narrow margin of the hyperechoic tissue (Figure 1A,B). During the whole study, the follow-up examinations revealed no changes to the prostatic appearance. The prostate gland in dogs from Group II (dogs with BPH) appeared as uniformly enlarged, and heterogenous with mostly coarse parenchymal texture, scattered hyperechoic foci, focal, less than 1 cm diameter, hypoechoic lesions (90%), or focal calcifications (10%) (Figure 1C–E). The initial appearance did not change throughout the study. Initially, a similar image of the prostate was noted in dogs from study Group III and IV (Figure 1C–E). Moreover, in all the investigated dogs the prostate gland was still symmetrical, with margins well differentiated from the surrounding tissues, and the inguinal lymph nodes were of normal size and presentation. In Group III from W8 onwards we observed some improvement in the heterogeneous appearance of the prostate parenchyma. In W16 decreased echo texture was noted in 93% of dogs, the gland became less coarse in 80% of patients, and more homogeneous in all investigated dogs (100% of patients), parenchymal cysts were not visible from W16. Whereas, in Group IV the echogenicity started to decrease as early as from D14 in 40% of the investigated dogs. Similarly, the image of parenchyma became more homogenous starting from D14, but this was noted only in 60% of the investigated dogs. The focal hypoechoic lesions disappeared in 60% of the dogs in W8 or W12 (Table 1, Figure 2 and Figure 3).

Prostate volume remained unchanged in Groups I and II throughout the study. Both OA and DA reduced the prostate volume. OA had a quicker effect (from D14) on prostate volume reduction compared to DA, but its effect lasted a shorter time (on average up to week 20), while DA reduced the prostate volume more slowly (W + 8), but the reduction level was higher, and persisted longer (>W + 24) (Table 2).

To reliably monitor the prostate volume in the study groups despite breed and size differences, the absolute values (cm^3^) were converted into percentages (%), assuming the D0 value as 100%. The changes noted during the study were analyzed in relation to this initial value, estimated as 100%. In Group III, the deslorelin acetate group, the period of most intensive reduction was observed later than in Group IV, but the prostate reached much smaller sizes (34% of the initial volume) at the end of the trial (W + 24). The significant reduction compared to Control Group II was first noted on D14 (*p* = 0.03) (Table 3), while compared to D0, in W + 8 prostate volume reached 47% of the initial size (*p* < 0.001) (Table 2). The prostate volume was the smallest in W + 24, at 34% of the initial volume. The rate of reduction was the highest between weeks 3 and 8 of the therapy (33% reduction), and 8 and 12 (7%), thereafter the prostate volume continued to decrease, but not significantly (Table 4). Whereas in Group IV the prostate volume decrease was noted sooner, starting from D7, compared to Control Group II (*p* = 0.01) (Table 3). A significant decrease compared to D0 was first noted on D14 (65% of the initial volume, *p* = 0.01) and remained at around 60% until W + 24, unfortunately then it started to increase and reached 84% of the D0 volume in W + 24 (Table 2). The most intense reduction occurred between the 1st and 2nd weeks of the study (Table 4).

### 3.2. Doppler Ultrasonography of the Prostatic Artery

We were only able to image the prostatic artery in short segments, due to its tortuosity [4]. Initially, the SPV and Vmean were higher in all dogs diagnosed with BPH, compared to Control Group I. The indices did not change in both Control Groups I and II, whereas in study Groups III and IV they decreased throughout the study. In Group III on D14 the SPV and Vmean were significantly lower than noted on D0 (SPV *p* = 0.007, Vmean *p* = 0.008) or in Control Group II (SPV *p* = 0.003; Vmean *p* = 0.006). On D21 the mean values were similar to Group I, and from the 8th week of the therapy decreased significantly compared to the healthy dogs in Group I. Whereas, in Group IV the SPV and Vmean were significantly lower, as early as on D7, compared to the D0 values (SPV *p* = 0.007; Vmean *p* = 0.008) or Control Group II (SPV *p* < 0.001, Vmean *p* = 0.05). The SPV and Vmean indices were similar in heathy dogs from Group I on D7, and remained similar throughout the study period. The highest reduction level of mean velocity was noted in Group III between D21 and W8. While in Group IV the lowest mean velocity was recorded on D21, and thereafter the mean SPV and Vmean increased, but not significantly (Figure 4). Only individual differences were recorded among groups regarding the RI value (Figure 5). The velocity of blood flow correlated positively with prostate volume in both study Groups III and IV (SPV: r > 0.52 *p* < 0.001, Vmean: r > 0.53 *p* < 0.001, and SPV: r > 0.45 *p* < 0.001, Vmean: r > 0.45 *p* < 0.001, respectively). There was no such correlation for RI indices in the study groups, while in Control Groups I and II no correlations between prostate volume and blood velocity were noted during the observation period (Figure 5, Table 5).

### 3.3. Testicles B-Mode Appearance

In all the investigated dogs the testes were echogenic, with homogenous medium echo texture and mediastinum imaged as a echogenic central linear structure. In Group II 20% of the dogs had a mild widening of the mediastinum, two dogs had isolated small cysts located in the testicular cranial plane, and four dogs had focal fibrosis in the testicular parenchyma. In this group (II) no changes were noted during the whole study period. In dogs from Group III, from W8 of the treatment, testicular parenchyma started to be less echoic, in W12 in 87% and in W20 in all of investigated dogs testicular parenchyma was described as hypoechoic, whereas the mediastinum first widened (W + 12), but then almost disappeared (W + 24). Moreover, the small fluid-filled cyst (<10 mm) found in one dog and the focal fibrosis imaged near the mediastinum in seven dogs on D0 were not visible form W16 onwards. In Group IV the focal changes and focal fibrosis noted on D0 in some dogs did not change in the following US examinations throughout the whole observation period. Similarly, the imaging of the testicular parenchyma remained unchanged in the following observations.

Testicular volume decreased significantly only in study Group III (deslorelin acetate group). The volume reduction compared to Control Group II was observed starting from W + 8 (*p* = 0.002), and compared to initial volume (D0) from W + 12, reaching up to 45% (*p* = 0.002). The testes were the smallest in W16 (35% of the initial D0 volume), and did not change till the end of the observation period. The most intense testicular volume reduction was noted between D21 and W12 after placing the Suprelorin^TM^ implant (Table 6, Figure 6).

### 3.4. Doppler Ultrasonography of the Testicular Artery

Generally, all the investigated dogs with BPH had lower values of mean SPV and Vmean than healthy dogs (Control Group I). Significantly lower flow velocities were noted on D0, D7, D14, W20, and W24 compared to Control Group I. Only in study Group III had the SPV and Vmean decreased from W8 of treatment compared to D0 and Control Group II values (SPV *p* < 0.001 and Vmean *p* < 0.001; SPV *p* < 0.001 and Vmean *p* < 0.001, respectively), and they continued to be lower till the end of the observation period. The highest flow velocity decrease was noted between D21 and W8. Moreover, in the last four examinations (W12, W16, W20, and W24) it was difficult to visualize the testicular vessels and correctly record the blood velocity, due to the progressive reduction in the blood flow and the narrowing of the testicular artery. In Group IV no differences were noted in SPV and Vmean throughout the study. Only individual variances were noted in the RI indices; lower values were noted in Control Group II than in Group I on D14 (*p* = 0.02) and in W + 20 (*p* = 0.01), and in Group IV in W8 of the study (*p* = 0.02), whereas in Group III, RI remained similar throughout the study (Figure 7 and Figure 8).

The velocity of blood flow in the testicular artery only correlated positively with testicular volume in study Group III (SPV: r > 0.55; *p* < 0.001; Vmean: r > 0.5; *p* < 0.001). In the remaining control (I, II) and treated (IV) groups no changes to blood flow and testicular volume were noted, hence no correlations were calculated. Furthermore, no correlation was found between RI and volume in any of the investigated or study groups (r > 0.18; *p* = 0.04).

## 4. Discussion

Ultrasound examination is one the most useful tools in diagnosis of prostate diseases, both in human and veterinary medicine. US is a fast, non-invasive, cheap, and usually easy to perform method in clinical practice [14,21]. As presented in this work, the regular US scanning of prostate and testicles during the study period, enabled us to observe even mild changes in echo texture or blood flow velocity, which indicated drug dependent structural changes of the given organs. The presented results confirm the clinical efficacy of ultrasound examination for diagnosis and progress monitoring of BPH in dogs.

In all of the investigated dogs with confirmed BPH, the US image of the prostate parenchyma was hyperechoic, coarse, and in most cases heterogenic, due to the presence of a large number of small, intraprostatic cysts (<10 mm), and less commonly, bigger (>10 mm) solitary cysts. In 9% of the investigated dogs with BPH we also noted paraprostatic cysts. Similar findings regarding the prostate parenchyma in case of BPH were reported by Russo et al. [14], and the presence of paraprostatic cysts was also described by other authors [1,22]. During the trial we noted no change to the US images of the prostate in dogs from Control Group I and II, whereas in Study Groups III and IV we observed very marked changes. In dogs receiving deslorelin acetate, with time the prostatic parenchyma became first normo- and later hypoechoic. These findings are very similar to those reported in surgically castrated dogs [23]. Furthermore, the small, intraprostatic cysts gradually resolved, leading to a more homogenous appearance of the parenchyma. Similar changes in echo texture were described by Goericke-Pesch et al. [24] with the use of azagly-nafarelin. While Jurczak et al. [25] reported the resolution of the intraprostatic cyst during deslorelin acetate treatment in all investigated dogs but one, with larger cysts (>58 mm and >115 mm) diagnosed, we observed cyst resolution in all the investigated dogs, however in our case the diameter of the largest cyst found did not exceed 40 mm. On the contrary, in group IV, receiving osaterone acetate, an improved, less echoic prostate parenchyma was noted only in 40% of dogs, and most of the cysts, especially those over 10 mm, remained unchanged during the whole observation period. Interestingly, in some of the investigated dogs receiving osaterone acetate, in week 20–24 of the study, the US appearance of the prostate parenchyma returned back to the images obtained before starting the treatment, suggesting the termination of the osaterone acetate action.

Both OA and DA reduced the prostate volume. The OA had a quicker effect on prostate volume reduction compared to DA, but its effect lasted a shorter time (on average up to 20 weeks), while DA reduced the prostate volume more slowly, but the reduction level was higher and remained longer, at least up to 36 weeks. This is consistent with differences in OA’s and DA’s methods of action. Deslorelin acetate, the GnRH agonist, first causes a marked increase of LH and FSH levels in the bloodstream, and later leads to pituitary desensitization, expressed by the internalization of the GnRH receptors, reduction of circulating LH and FSH, and inhibition of testosterone production in testicular tissue [11]. Low testosterone levels do not stimulate prostate cells in BPH individuals, and overtime lead to the gland atrophy, similar to that seen after surgical castration. However, these changes to the prostatic tissue are gradual and clinically visible after a few weeks of the therapy. Osaterone acetate is a steroid, chemically related to progesterone, which affects the testosterone uptake by the prostatic tissue, reducing, but not blocking it fully [9]. That is why the volume reduction, although present, is not as significant as for GnRH analogue (which completely stops testosterone production), but occurs much faster from the treatment onset [11]. The gland echogenicity remains almost unchanged with the use of OA. Similar results regarding volume reduction were published by Goericke-Pesch et al. [24] with the use of azagly-nafarelin. Here, the prostate volume was reduced by up to 64% in week 26 of the study. Prostate volume reduction was also observed by Ponglowhapan and Lohachit [26], and Jurczak et al. [25], with the use of GnRH agonist, but they did not published the exact numbers. While Polisca et al. [15] observed a significant volume reduction earlier than in our study, from day 52 using deslorelin acetate. Most probably this was related to the breeds of dog selected for the study groups in both studies. Polisca et al. [15] investigated only German shepherd dogs, of similar age and weight, whereas our investigations included dogs of various breeds, ages and weights. The obtained results in the current study, together with the results presented by Niżański et al [18] suggest that in cases with severe BPH symptoms it may be beneficial to initiate the therapy using both OA and DA together, to achieve a quick clinical response (OA) which lasts longer (DA), compared to the individual use of these drugs, but this needs further research.

Doppler images of the prostate were first published in humans by Neumaier et al. [27], who identified vascularity of healthy prostate gland, and later Tsuru et al. [20] described the blood flow in BPH affected glands. In veterinary medicine the first report on Doppler sonography of canine prostate and testes was published in 2001 by Guenzel-Apel et al. [12], and concerned healthy dogs and dogs with BPH. To date, there has only been one publication on Doppler ultrasound characteristics in the prostate of dogs receiving pharmacological treatment for BPH (deslorelin acetate) [15]. Our results confirmed the overall higher blood flow velocity in the prostatic artery of dogs suffering from BPH. The obtained mean velocities in healthy dogs (Group I) and dogs with BPH (Group II) were similar to data published by Guenzel-Apel et al. [12]; however, we noted a lower standard deviation in our results. In our study, in the OA group the mean velocity values decreased sooner (from day 7) than in the DA group (from week 12). However, in the OA group the noted mean velocity was only at a level similar to that noted in healthy dogs, whereas in the DA group, from week 5 of the therapy, the mean velocities were significantly lower compared to the healthy dogs. The significant velocity reduction in dogs receiving DA started earlier, from day 14 (SPV—27.57 cm/s, Vmean—8.27 cm/s) in our observations, compared to the study of Polisca et al. [14], who noted a similar reduction around 8 days later. On the other hand, we noted the lowest values (mean SPV—17.52 cm/s, Vmean—5.5 cm/s) in week 16, whereas for Polisca et al. [14] they were around week 12. As stated previously, this could be related to the different inclusion criteria for dogs between the studies. Moreover, we performed re-examinations every 30 days, whereas Polisca et al. [15] re-examined every 15 days, which could also have affected the obtained results. It is worth noticing that our results clearly indicate that DA reduced the blood flow in the prostatic artery during the whole therapeutic period, as guaranteed by the manufacturer, with the peak reduction between day 21 and week 5 of use. Interestingly, despite the widespread use of OA in treatment of BPH, and many studies on its influence on the size of the gland [10,20,28], there are no reports concerning blood flow changes in the prostatic arteries while using this medication. Our results showed a significant slowdown of the prostatic blood flow registered in the 1st week of the therapy (mean SPV in D0—34.36 cm/s, vs. D7 - 26,16 cm/s), but thereafter, the average flow rates reduced only slightly (SPV up to 20 cm/s) in the following examinations, and were similar to values described in healthy dogs [12]. The observed positive correlation between blood flow rates and prostate volume noted in study groups III and IV suggests that the gland vascular system is the first one affected by the reduced androgen concentrations. Particularly, in both study groups the flow indices reduced much earlier than a significant reduction in prostate volume was noticed. Some authors have already proposed that in the pathogenesis of BPH, the regulation of the prostate blood flow is a primary target of androgen action [29]. During treatment, the sonographic recovery of the prostate parenchyma, occurs secondarily to the regression of the prostate vascular system, through the cell death mediated by tissue ischemia/hypoxia [30]. The RI, a very reliable factor in human medicine for the differentiation between BPH and healthy patients, seemed to be unsuitable for BPH diagnosis in dogs according to our results, and the results published by Guenzel-Apel et al. [12] and Polisca et al. [15]. Since the RI measures the peripheral resistance, the higher values noted during prostatic enlargement in humans, are most probably associated with the elevated intraprostatic pressure, which compresses the blood vessels. This is contrary to what happens in dogs, as the mechanism of action is different, the prostate gland enlarges sideways and does not constrict the prostatic vasculature [31,32].

The changes of testicular parenchymal echogenity and testicular volume were observed only in dogs receiving DA, and were similar to those observed with the use of azagly-nafarelin published by Goericke-Pesch [24]. The remaining control groups I and II, and study group IV treated with OA, did not show any significant sonographic and volumetric changes of the testes. This is consistent with the results of Murakoshi et al. [33], who did not find any difference in Leydig cell appearance or seminiferous tubules in dogs treated with OA. On the contrary, Junaidi et al. [17] proved that DA induced a progressive atrophy of the seminiferous tubules, starting form day 16, which explains the sonographic changes observed in our investigations.

In veterinary medicine, the use of Doppler sonography for testicular blood flow has already been published in horses [34], dogs [12,35] and alpacas [36]. Guenzel-Apel et al. [12] compared hemodynamic changes between healthy testicles and testicles with neoplastic processes in dogs, and England et al. [34] looked for correlations between Doppler ultrasound examination and future semen quality. In our study the majority of the SPV and Vmean values observed in control groups I and II, and group IV receiving OA, were similar to values described in healthy dogs by Guenzel-Apel et al. [12]. Only in the DA group, did the testicular blood flow decrease during the trial, and it was accompanied by testicular involution, suggesting mutual testicular atrophy and a reduction of the testicular artery diameter related to deslorelin acetate’s mode of action. In dogs receiving OA, testicular blood flow did not change, which tends to show that osaterone acetate has no direct effect on the gonadal function in dogs. In humans, Biagiotti et al. [37] showed the relationship between Doppler studies of testicular flow and spermatogenesis. Infertile men had significantly lower SPV and RI values. Moreover, the RI was higher in prepubertal boys [38], in men older than 51 years [39], and during testicular inflammation [40]. In alpacas with reduced fertility, significantly lower SPV values were found, however RI did not differ between fertile and infertile individuals [35]. While in stallions, significantly higher RI values were found in older individuals, over 15 years of age [34]. To date, no such relationship has been demonstrated in dogs. England et al. found no relationship between ultrasound parameters and future, total sperm production [35]. Additionally, no significant changes to RI were reported for spermatic cord torsion in dogs [41]. The lack of change in RI parameters in dogs, might be associated with the scanning technique [34], or patient weight and testicular sizes. Paltiel et al. [38] reported a higher RI in individuals with bigger testicular volume. Unfortunately, the analysis of the obtained RI values was not easy, as we observed in our study. We recorded occasional differences in RI index among the investigated groups, but we failed to observe any regularity.

In clinical practice, the most prevalent (and often the first line) medical therapy for dogs suffering from BPH attributed symptoms, is the surgical castration. However, recently more and more owners prefer to avoid surgery, and choose less invasive methods. It is worth noticing that in some cases it might be beneficial to initiate pharmacological therapy using both OA and DA together, to obtain quick prostate reduction with the use of OA, and a longer lasting effect due to the DA action.

## 5. Conclusions

B-mode and Doppler ultrasonography should become the first screening tool to investigate prostate and testicular conditions. Primarily, B-mode US allows the depiction of anatomy, echogenicity, and size. However, alone it would not be adequate for the evaluation of a condition’s progress, or the therapeutic effectiveness of a drug. Whereas, color Doppler determines the nature of blood flow and vascularity, differentiating areas of high and low perfusion, and thus greatly improving the monitoring accuracy.

The present study proves the usefulness of B-mode and Doppler imaging techniques for the diagnosis and progress assessment in dogs suffering from BPH. The blood flow kinetics (mainly SPV) demonstrated a time association between the blood flow changes registered in the prostatic artery, and the subsequent volumetric and sonographic improvement of the prostate parenchyma. The flow indices were reduced much earlier than a significant reduction in prostate volume was noticed, suggesting that the regulation of the prostate blood flow is a primary target of androgen action. Consequently, it could be presumed that the sonographic recovery of the prostate parenchyma, occurs secondarily to the regression of the prostate vascular system. Moreover, the comparison of both investigated medications (antiandrogen, i.e., osaterone acetate, and deslorelin acetate, used for medical castration) showed some differences in the observed sonographic improvement. Deslorelin acetate reduced prostate volume more slowly, but the reduction level was higher, and its effect lasted longer (at least up to 36 weeks) than for osaterone acetate, which had a quicker effect on the prostate, but its effect lasted a shorter time (on average up to 20 weeks).

BPH in dogs is still a challenging problem in clinical practice. The recent diagnostic imaging advances allow for a more profound evaluation and understanding of the underlying mechanisms responsible for clinical improvement during benign prostatic hyperplasia treatment. The reported US changes in the prostate noted with the use of pharmacological components should also be compared to the changes occurring in dogs in which surgical castration was chosen as the BPH treatment, to see the difference between the presence or absence of gonadal steroidal feedback on the dynamics of changes in the prostate gland parenchyma. Hence, further research is needed to develop and characterize standard reference values for BPH in dogs, allowing better accuracy in ultrasonographic diagnostics and progress assessment.

## Figures and Tables

**Figure 1 animals-10-02379-f001:**
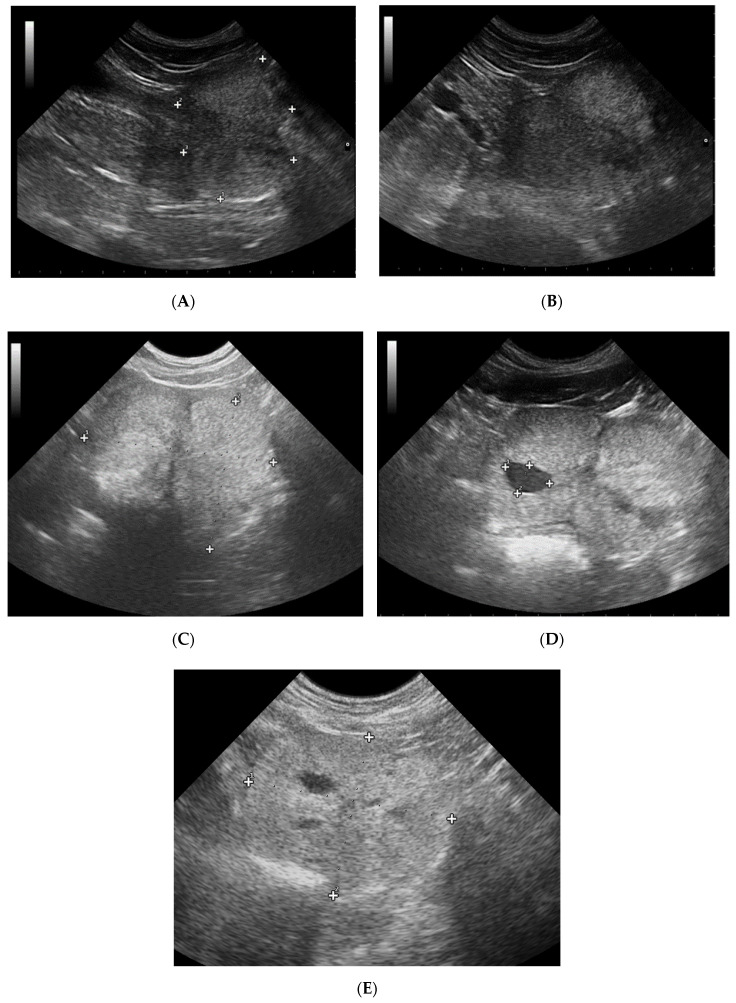
B-mode imaging of a normal prostate, and the most common appearance in dogs suffering from BPH. (**A**)—normal prostate in transverse view, the thin echogenic capsular margins are indicated (+); (**B**)—normal prostate in sagittal view; (**C**)—Benign prostatic hyperplasia, prostate gland in sagittal view, hyperechoic parenchyma and mild enlargement; (**D**)—BPH: large cyst; (**E**)—BPH: small anechoic cysts in parenchyma.

**Figure 2 animals-10-02379-f002:**
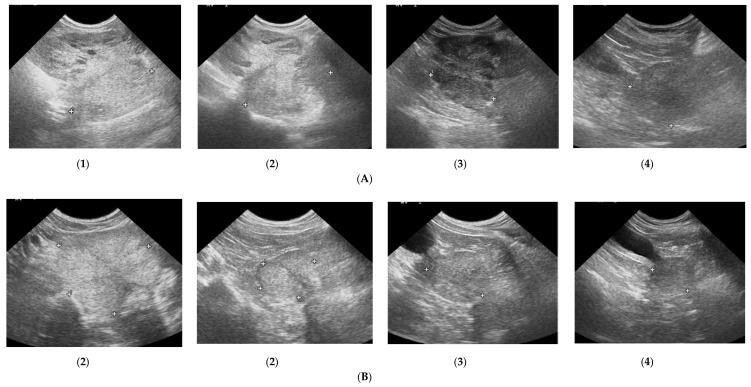
Changes in B-mode prostate imaging in dogs receiving deslorelin acetate, visible gradual involution of multiple cysts, and changes in echogenicity of the prostatic parenchyma. Letters (**A**) and (**B**) correspond to a single patient in the following examinations on days: (**1**)—day 0; (**2**)—week 8; (**3**)—week 16, and (**4**)—week 24. (**A**)—dog with large and multiple parenchymal cysts (>10 mm), sagittal view, right lobe indicated in pictures; (**B**)—dog with small and multiple parenchymal cysts (<10 mm), transverse view, the prostate gland indicated in pictures.

**Figure 3 animals-10-02379-f003:**
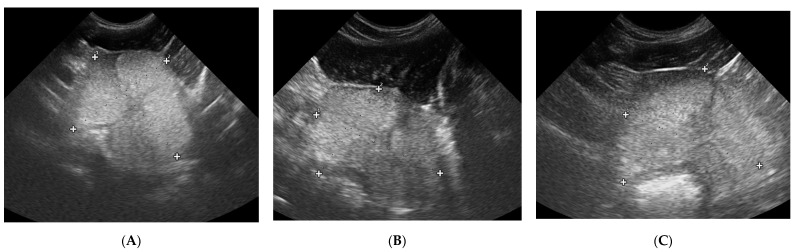
The B-mode prostate imaging in a dog receiving osaterone acetate, on days: (**A**)—day 0; (**B**)—week 8; (**C**)—week 16; transverse view, the prostate gland is indicated in pictures.

**Figure 4 animals-10-02379-f004:**
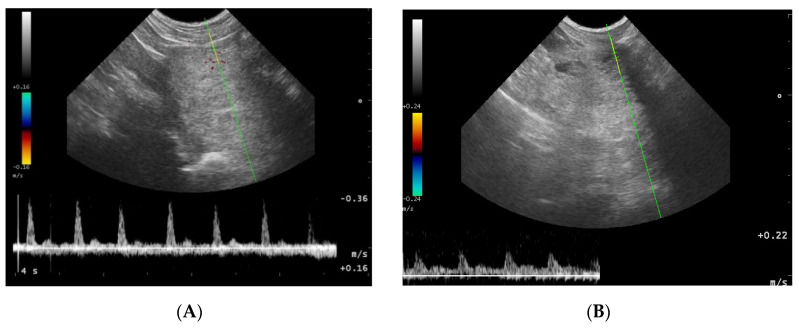
Blood flow patterns, using the duplex technique, of the prostatic artery. (**A**)—normal pattern in lateral approach; (**B**)—normal pattern in subcapsular approach; (**C**)—blood flow pattern in BPH, the flow towards probe, waveform manually traced; red arrow—systolic peak velocity, parabolic flow profile, no spectral window visible and the diastolic peak without flow reversal, a wave with a higher flow velocity visible in diastole—green arrow.

**Figure 5 animals-10-02379-f005:**
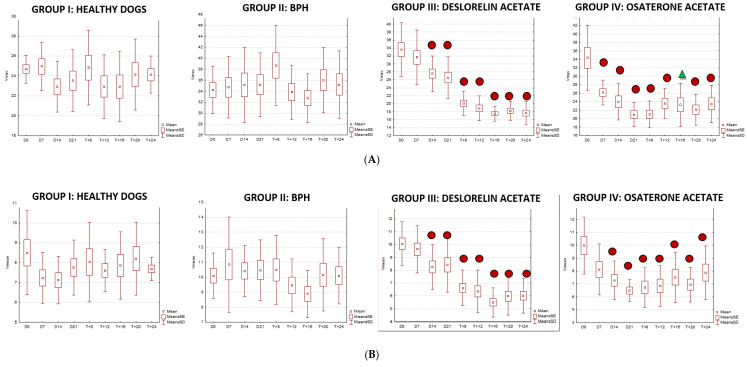
Changes of blood flow velocity ((**A**)—systolic peak velocity, SPV, cm/s; (**B**)—mean velocity, Vmean, cm/s; (**C**)—resistive index, RI) in the prostatic arteries compared to Day 0 (D0) values in the investigated groups of dogs during the 24 weeks of observation (mean, SD, SE).

**Figure 6 animals-10-02379-f006:**
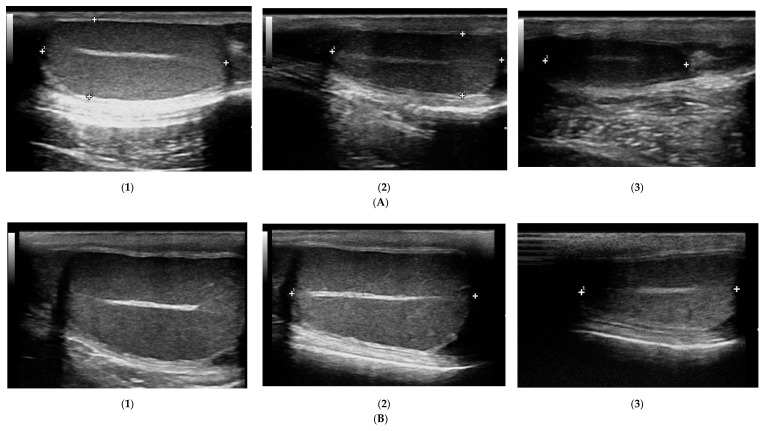
Changes in B-mode testicular imaging in dogs receiving deslorelin acetate. Visible progressive disappearance of mediastinum and gradual lowering of tissue echogenicity. (**A**,**B**)—patients form Study Group III, examined on (**1**)—day 0; (**2**)—week 8; (**3**)—week 24.

**Figure 7 animals-10-02379-f007:**
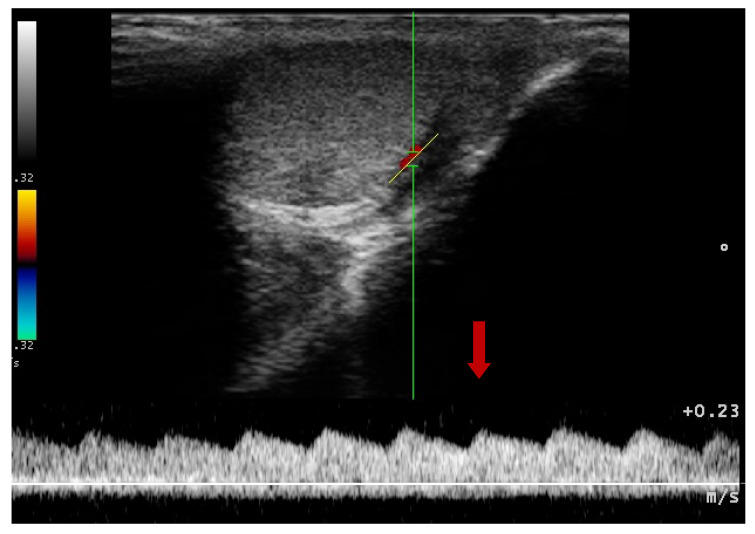
Blood flow pattern using a duplex technique in the testicular artery. Flow towards probe, red arrow—systolic peak velocity, wide parabolic flow profile, no spectral window visible, gradual decrease of velocity in diastole.

**Figure 8 animals-10-02379-f008:**
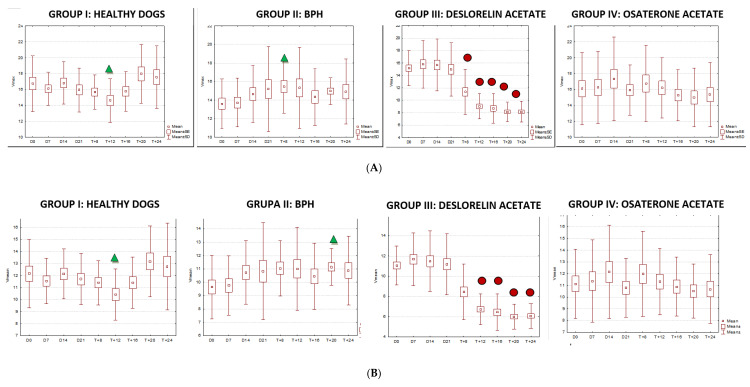
The changes of the blood flow velocity ((**A**)—systolic peak velocity, SPV, cm/s; (**B**)—mean velocity, Vmean, cm/s; (**C**)—resistive index, RI) in the testicular arteries compared to day 0 (D0) values, in the investigated groups of dogs, during the 24 weeks of the study (mean, SD, SE).

**Table 1 animals-10-02379-t001:** Subjective assessment (0%—none; 100%—present in the entire image) of prostate gland parenchyma in B-mode imaging after Russo [14], in Control Groups I and II, and Study Groups III and IV during 24 weeks of observation.

	**Control Group I—Healthy Dogs**											**Control Group II—Dogs with BPH**										
	**Background Echotexture**			**Parenchymal Stippling**			**General Apperance**		**Focal Changes**			**Background Echotexture**			**Parenchymal Stippling**			**General Apperance**		**Focal Changes**		
	**N**	**HI**	**HY**	**R**	**C**	**I**	**HO**	**HE**	**CY**	**M**	**fHY**	**N**	**HI**	**HY**	**R**	**C**	**I**	**HO**	**HE**	**CY**	**M**	**fHY**
**D0**	100%	0	0	100%	0	0	100%	0	0	0	0	0	100%	0	30%	70%	0	0	100%	90%	10%	0
**D7**	100%	0	0	100%	0	0	100%	0	0	0	0	0	100%	0	30%	70%	0	0	100%	90%	10%	0
**D14**	100%	0	0	100%	0	0	100%	0	0	0	0	0	100%	0	30%	70%	0	0	100%	90%	10%	0
**D21**	100%	0	0	100%	0	0	100%	0	0	0	0	0	100%	0	30%	70%	0	0	100%	90%	10%	0
**W+8**	100%	0	0	100%	0	0	100%	0	0	0	0	0	100%	0	30%	70%	0	0	100%	90%	10%	0
**W+12**	100%	0	0	100%	0	0	100%	0	0	0	0	0	100%	0	30%	70%	0	0	100%	90%	10%	0
**W+16**	100%	0	0	100%	0	0	100%	0	0	0	0	0	100%	0	30%	70%	0	0	100%	90%	10%	0
**W+20**	100%	0	0	100%	0	0	100%	0	0	0	0	0	100%	0	30%	70%	0	0	100%	90%	10%	0
**W+24**	100%	0	0	100%	0	0	100%	0	0	0	0	0	100%	0	30%	70%	0	0	100%	90%	10%	0
	**Experimental Group III—Deslorelin Acetate**											**Experimental Group IV—Osaterone Acetate**										
	**Background echotexture**			**Parenchymal stippling**			**General apperance**		**Focal changes**			**Background echotexture**			**Parenchymal stippling**			**General apperance**		**Focal changes**		
	**N**	**HI**	**HY**	**R**	**C**	**I**	**HO**	**HE**	**CY**	**M**	**fHY**	**N**	**HI**	**HY**	**R**	**C**	**I**	**HO**	**HE**	**CY**	**M**	**fHY**
**D0**	0	100%	0	7%	93%	0	0	100%	100%	0	0	0	100%	0	0	100%	0	0	100%	100%	0	0
**D7**	0	100%	0	20%	80%	0	0	100%	100%	0	0	0	100%	0	0	100%	0	0	100%	100%	0	0
**D14**	0	100%	0	20%	80%	0	0	100%	100%	0	0	20%	80%	0	10%	90%	0	20%	80%	80%	0	0
**D21**	0	100%	0	27%	73%	0	0	100%	100%	0	0	40%	60%	0	50%	50%	0	50%	50%	50%	0	0
**W+8**	40%	33%	27%	87%	6.5%	6.5%	77%	33%	33%	0	0	40%	60%	0	50%	50%	0	60%	40%	40%	0	0
**W+12**	40%	0	60%	87%	0	13%	93%	7%	7%	0	0	40%	60%	0	50%	50%	0	60%	40%	40%	0	0
**W+16**	7%	0	93%	20%	0	80%	100%	0	0	0	0	30%	70%	0	30%	70%	0	50%	50%	50%	0	0
**W+20**	7%	0	93%	20%	0	80%	100%	0	0	0	0	20%	80%	0	30%	70%	0	50%	50%	50%	0	0
**W+24**	7%	0	93%	20%	0	80%	100%	0	0	0	0	30%	70%	0	30%	70%	0	30%	70%	0	0	0

N—normal, H+—hyperechoic, H-—hypoechoic; R—regular, I—increased, C—coarse; HO—homogenous, HE—heterogenous; CY—cysts, M—mineralized opacities, fHY—focal hypoechoic lesions.

**Table 2 animals-10-02379-t002:** Prostate volume (v, cm^3^) in control I and II, and treated III and IV groups in the following examinations during the study period. Statistical analysis in relation to **day 0** (D0–D21, W + 8–W + 24, mean, SD, range).

Group	D0	D7	D14	D21	W + 8	W + 12	W + 16	W + 20	W + 24
I									
Mean	15.58	16.06	14.82	14.84	16.12	15.77	16.26	16.87	17.5
SD	3.48	3.53	2.97	2.65	2.46	2.8	3.62	3.42	3.63
Range	11.55–20.8	12.46–21.61	11.35–20.34	11.49–19.35	12.61–19.09	12.55–19.21	12.03–22.11	12.41–23.88	13.53–24.6
II									
Mean	18.26	19.04	19.1	19.24	19.05	20.67	21.47	21.93	20.44
SD	4.1	4.72	4.23	4.7	4	5.83	6.63	7.01	6.5
Range	12.82–23.32	12.4–26.7	12.64–23.84	11.88–26.91	13.47–24.63	13.93–29.84	11.97–36.55	12.24–34.5	11.75–31.52
III									
Mean	39.39	39.11	32.05	28.35	14.65 *	11.62 *	10.7 *	9.6 *	9.26 *
SD	31.26	26.15	19.16	20.62	7.9	5.4	3.41	1.72	1.47
Range	11.13–115.91	16.01–106.49	12.88–80.79	11.76–92.57	8.44–40.94	8.5–30.58	7.82–21.83	7.88–14.52	7.7–12.61
IV									
Mean	24.07	20.74	14.7 ^&^	13.21 *	13.68 *	13.65 *	13.79 *	14.7 ^&^	18.57
SD	9.82	6.96	4.1	2.8	4.09	3.72	4.12	5.3	9.55
Range	10.29–38.48	10.91–32.7	8.42–22.5	8.3–16.78	8.48–19.83	8.73–21.53	8–19.5	8.5–25.11	7.61–38.3

^&^*p* < 0.05 in relation to D0; * *p*,0.01 in relation to D0; in italics—compared using Mann–Whitney test, normal fonts—T-student test.

**Table 3 animals-10-02379-t003:** Comparison of prostate volume changes between control and treated groups. To reliably monitor the prostate volume in the study groups despite breed and sizes differences, the absolute values (cm^3^) were converted into percentages (%). Prostate volume presented as percentage (%), where the initial (D0) volume was considered 100%.

Groups Compared	D7	D14	D21	W + 8	W + 12	W + 16	W + 20	W + 24
I C–II C	0.92020	0.04165 ^&^	0.18131	*0.62421*	*0.01137* ^&^	*0.08641*	0.14046	0.95508
II C–III T	*0.91166*	0.03259 ^&^	0.00279 *	*<0.00001 **	*0.00003 **	0.00003 *	<0.00001 *	<0.00001 *
II C–IV T	0.01195 ^&^	<0.00001 *	<0.00001 *	<0.00001 *	*0.00016 **	*0.00016 **	*0.00016 **	0.06155
III T–IV T	*0.02650* ^&^	0.00116 *	0.01486 ^&^	0.07162	0.00616 *	0.00141 *	*0.00072 **	0.00024 *

Significance level between groups: probability (*p*). Abbreviations: Control Group I (I C), Control Group II (II C), Study Group III (III T), Study Group IV (IV T). ^&^
*p* < 0.05; * *p*,0.01; in italics—compared using Mann–Whitney test, normal fonts—T-student test.

**Table 4 animals-10-02379-t004:** The prostate volume changes (V, cm^3^) in both Study Groups III and IV. Significance level in relation to D0: probability (*p*).

Groups	D7	D14	D21	W + 8	W + 12	W + 16	W + 20	W + 24
III	*0.85193*	*0.41862*	*0.37251*	*0.00264 **	*0.04882 **	*0.46792*	*0.41862*	*0.46792*
IV	0.39270	0.02959 *	0.35476	0.76735	0.98397	0.93458	0.67268	0.27672

* *p* < 0.05; in italics—compared using Mann-Whitney test, normal fonts—T-student test.

**Table 5 animals-10-02379-t005:** Blood flow indices (systolic peak velocity—SPV, cm/s; mean velocity—Vmean, cm/s) in the prostatic arteries compared between control (Group I and II) and study (III and IV) groups during the 24 weeks of observation (mean, SD, SE).

Group	D0		D7		D14		D21		W+8		W+12		W+16		W+20		W+24	
	SPV	Vmean	SPV	Vmean	SPV	Vmean	SPV	Vmean	SPV	Vmean	SPV	Vmean	SPV	Vmean	SPV	Vmean	SPV	Vmean
**I**																		
**Mean**	24.68 ^a^	8.5	24.95 ^a^	7.23 ^a^	22.91 ^a^	7.12 ^a^	23.51 ^a^	7.75 ^a^	24.82 ^a^	8.04 ^a^	22.93 ^a^	7.60 ^a^	22.94 ^a^	7.86 ^a^	24.13 ^a^	8.21 ^a^	24.82 ^a^	8.04 ^a^
**SD**	1.4	2.12	2.44	1.29	2.56	1.19	3.11	1.39	3.743	2.01	3.19	1.055	3.53	1.69	3.58	1.84	3.74	2.01
**SE**	23.1–27.28	6.34–13.6	21.3–29.53	6.02–10.53	18.65–27.5	5.78–9.24	16.18–26.43	5.44–10.6	30.53–18.03	3.7–11.03	17.95–25.34	6.05–9.74	15.9–27.95	5.37–11.53	19.48–32.27	5.48–11.47	20.98–26.02	6.7–8.48
**II**																		
**Mean**	34.22 ^b^	10.11	34.71 ^b^	10.84 ^b^	35.16 ^b^	10.43 ^b^	35.17 ^b^	10.48 ^b^	38.69 ^b^	10.49 ^b^	33.82 ^b^	9.46 ^b^	32.74 ^b^	8.88 ^a^	36.00 ^b^	10.16 ^b^	35.21 ^b^	10.10 ^b^
**SD**	4.34	1.51	5.60	3.18	6.85	1.7	5.81	2.014	7.29	2.3	4.97	1.756	4.41	1.57	5.920	2.41	6.14	1.87
**SE**	24.88–38.88	7.13–11.76	25.78–44.38	5.58–15.23	25.54–45.62	7.9–12.94	27.9–39.6	7.1–13.73	31.48–52.72	7.84–15.84	27.9–46.23	8.18–13.9	29–44.7	6.13–11.26	25.88–47.47	5.85–12.37	26.98–45.7	7.7–14.35
**III**																		
**Mean**	33.65 ^cb^	10.06	31.66 ^cb^	9.64 ^cb^	27.57 ^c^	8.27 ^a^	26.56 ^ac^	8.42 ^a^	20.13 ^c^	6.63 ^a^	18.85 ^c^	6.35 ^c^	17.52 ^c^	5.50 ^b^	18.24 ^c^	5.97 ^c^	17.7 ^c^	6.02 ^c^
**SD**	6.83	1.70	6.87	1.85	4.47	1.75	5.23	2.14	3.05	1.38	3.15	1.66	1.85	1.14	2.39	1.454	2.91	1.36
**SE**	21.22–45.37	7–13.37	19.98–45.37	6.4–12.73	20.9–34.03	5.78–11.73	18.55–33.87	5.84–13.76	16.3–27.8	5.08–9.36	14.7–25.83	4.52–11.28	15.24–19.05	3.22–8.22	14.62–21.93	3.84–9.35	14.36–25.08	3.73–7.64
**IV**																		
**Mean**	34.36 ^dbc^	9.98	26.16 ^a^	8.12 ^a^	23.98 ^ac^	7.28 ^a^	20.99 ^ad^	6.5 ^c^	21.05 ^dc^	6.74 ^a^	23.59 ^a^	6.85 ^ac^	23.25 ^a^	7.52 ^a^	22.14 ^a^	6.95 ^ac^	23.48 ^a^	7.87 ^a^
**SD**	7.66	2.21	2.91	1.98	4.32	1.5	2.91	0.86	3.13	1.57	3.55	1.58	5.13	1.94	3.61	1.34	4.43	2.09
**SE**	17.8–42.87	7.5–12.23	22.7–31.29	6.3–11.57	17.27–31.96	6.13–10.25	16.36–26.8	5.36–8.26	14.87–25.3	3.83–8.92	18.96–25.96	5.28–9.98	17.92–31.13	5.34–10.88	17.46–29.4	5.7–10.27	16–29.92	3.7–11.22

Values in columns with different letters differ (*p* < 0.05).

**Table 6 animals-10-02379-t006:** Testicular volume (v, cm^3^) in control I and II, and study III and IV groups in the following examinations during the study. Statistical analysis in relation to **day 0** (D0–D21, W + 8–W + 24, mean, SD, range).

Group	D0	D7	D14	D21	W + 8	W + 12	W + 16	W + 20	W + 24
I									
Mean	8.43	*10.51*	8.01	*7.86*	*8.62*	8.24	8.24	8.25	8.45
SD	3.49	*7.59*	3.02	*3.03*	*3.23*	2.49	3.43	3.2	3.34
Range	5.06–15.07	*5.33–29.3*	5.0–13.62	*5.1–13.68*	*5.85–14.61*	5.68–13.1	4.98–14.12	4.92–13.14	5.28–13.65
II									
Mean	4.47	4.71	4.7	4.66	*4.98*	4.87	4.99	4.9	5.27
SD	2.2	2.81	2.69	2.89	*3.38*	2.8	3.04	2.7	3.15
Range	1.56–8.24	1.65–10.29	1.25–9.51	0.85–10.42	*0.85–12.1*	1.05–10.41	1.4–11.45	0.98–9.82	1.4–12.01
III									
Mean	10.22	10.75	10.12	9.23	6.97	*4.92 **	3.69 *	3.85 *	3.68 *
SD	4.93	5.67	4.72	4.18	4.3	*3.85*	2.18	2.5	2.26
Range	3.58–22.8	3.6–24.85	3.6–20.12	3.18–17.79	1.41–15.38	*1.2–17.07*	0.84–8.9	0.88–9.28	0.79–8.4
IV									
Mean	11.61	10.43	10.53	10.68	10.47	11.25	10.6	10.23	11.28
SD	7.03	5.95	6.53	6.37	6.44	6.94	6.58	6.05	7.78
Range	2.02–24.22	2.43–20.42	2.4–23.8	2.83–21.15	2.1–20.23	2.7–17.63	2.02–23.07	2.0–21.04	1.93–21.42

* *p*,0.01 in relation to D0; in italics—compared using Mann–Whitney test, normal fonts—T-student test.

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
