# Peer review of "B-Mode and Doppler Ultrasonographic Findings of Prostate Gland and Testes in Dogs Receiving Deslorelin Acetate or Osaterone Acetate"

_animals, 2020, doi:10.3390/ani10122379_

Round 1
Reviewer 1 Report
Dear authors, thank you for the modifications that you have accepted to make following my suggestions. I think your study and your manuscript are really interesting and useful for the veterinary community, and it deserves to be published.
I don't completely agree with your justifications concerning group II (dogs with BPH but no treatment administered) - but it is not very important. I still don't understand why dogs owners or breeders who have a dog showing clinical signs of BPH (hemospermia, sperm defects etc...) may not want to treat their dogs. But this may be due to different attitudes of owners in your country and in mine. Because your manuscript is really good, I don't want to block in anyway its publication, therefore I am really happy to accept the publication in the present form. Congratulations.
Best regards.
Reviewer 2 Report
The paper now reads fairly well although the English style could be improved. There is an excessive use of commas and an excessive (and often inappropriate) use of the article "the". On pages 25-26 he font is different from the rest of the paper - this should be changed.
The results are interesting and well presented, overall it is a good paper.
This manuscript is a resubmission of an earlier submission. The following is a list of the peer review reports and author responses from that submission.
Round 1
Reviewer 1 Report
Dear authors, I am a little bit annoyed making comments on your manuscript. It is exceptionnally well written, well described, and brings very interesting new knowledge to the veterinary community. I enjoyed reading it very well. It deserves to be published, but I think that somme details and comments should be added.
However, there are some things that are lacking and which distrub me. Your enthusiasm lead you nearly to claim a "treatment" of BPH (i.e. lines 598-599) while you are only describing the effect of DA and OA on the ultrasonographic appearence of the prostate. I think one of the weak point of your manuscript is to remain completely silent about the clinical symptoms presented by the dogs in groups II, III and IV. Although you state (line 643) that all dogs had "clinical symptoms of BPH", you never describe them, nor indicate the clinical improvement obtained during the study. In "materials and methods"(paragraph 2.7. Indications for inclusion), you just mention the B-mode appearence of the prostate and the US-FNA but the clinical symptoms are completely forgotten. It is still a debate among specialist to consider if a dog having on heterogeneous prostate on ultrasound but no clinical signs has to be treated and can be considered as having a "clinical BPH". And because you don't mention the clinical symptoms, you don't correlate the clinical improvement with the modifications observed on ultrasound, which would have been very interesting. I understand that it was not your initial goal, but I find essential that you speak about the clinical aspects of the dogs enrolled in the study. It should be added and discussed.
Another frustrating point is that you didn't decide to add one group of BPH dogs that were surgically castrated. It should be also discussed.
Now I am going to give a few statements line by line.
- lines 59-60: you state that to make a definitive diagnosis of prostate pathology a transabdominal prostatic biopsy is necessary. I don't fully agree. Of course, a prostatic biopsy appears as the gold standard, but FNA (which has quite a high correlation with biopsy) or even prostatic massage are often conclusive. Furthermore, I think there is a confusion in your mind between "FNA" and biopsy", as in "materials and methods" (line 164), you write that you were performing US-guided FNA, and in lines 167 and 169 you write "after prostate biopsy". I think this is contradictory and should be clarified;
- line lines 80-82: "it contains a steroidal androgen which... and inhibiting the action of 5a-reductase": please give a fundamental reference of this statement, as the exact mecanism of action of osaterone acetate is often unclear;
- line 106: table 1 (breeds) does not exist in what I received to review, as in what I had to review table 1 reports about the "subjective assessment of prostate gland parenchyma". Please add this table to your manuscript;
- lines 109-111: group II (dogs with BPH but "no treatment administered", not even a placebo) raises an ethical concern to me. Either they were not having clinical signs - and therefore what you write on line 643 is wrong - or they were not clinically diseased, and it is the point I already spoke about. Anyway, if they had clinical signs, how did the owners accepted that their dogs were not treated ? Did they sign a written agreement ? What were the motivated explanations of your ethical committee to accept this ? And what about if these dogs had clinical signs appearing during the study or, if they had already clinical signs at the beginning, if these clinical signs had increased. Would they have been excluded from the study and fully treated ? You don't say anthing about it. Please add more explanations.
- lines 219-220: "whereas, in group IV ... 40% of the investigated dogs": this sentence is unclear, please could you rephrase it ? Espacially I don't understand what you mean when writing "it did not exceed more than 40% of the investigated dogs".
- figure 2 is interesting. It could be useful to add a similar figure (figure 2bis ?) with photos of the action of osaterone acetate.
- lines 494-495: you state that the mediastinum testis disappeared at W+24, but on the photo of the figure 5 (C), it is not evident.
- lines 500-505: a comment on the findings of testicularB-mode appearence of group IV during treatment is lacking.
- lines 552-554: "the RI indices were lower in group II ... whereas in group IV RI indices were lower in W8 of the study (p=0.02)": this is an interesting finding which should be discussed in "discussion", which potential explanations.
- lines 598-599: already said: clinical data are lacking to claim a "treatment".
- line 643: already said: clinical symptoms are not mentioned in your manuscript.
- lines 644- 645: I agree that - following you results - it seem interesting to combine OA and DA for the treatment of BPH but because no study has been done (it could have been another group of you study), I would prefer - line 644 - that you write: "it may be beneficial" rather than "it would be beneficial".
- lines 704-705: I don't agree with the term "confirms". I would rather write "it tends to show" or a similar statement, not so affirmative.
- line 720: already said: a group of surgically castrated dogs with BPH is lacking in your study and your choice not to include it should be discussed in "discussion".
Thank you very much. Best regards.
Author Response
Authors would like to thank the reviewer for comments and suggestions which helped to improve the MS. All suggested changes have been addressed as follows:
Dear authors, I am a little bit annoyed making comments on your manuscript. It is exceptionnally well written, well described, and brings very interesting new knowledge to the veterinary community. I enjoyed reading it very well. It deserves to be published, but I think that somme details and comments should be added.
However, there are some things that are lacking and which distrub me. Your enthusiasm lead you nearly to claim a "treatment" of BPH (i.e. lines 598-599) while you are only describing the effect of DA and OA on the ultrasonographic appearence of the prostate. I think one of the weak point of your manuscript is to remain completely silent about the clinical symptoms presented by the dogs in groups II, III and IV. Although you state (line 643) that all dogs had "clinical symptoms of BPH", you never describe them, nor indicate the clinical improvement obtained during the study. In "materials and methods"(paragraph 2.7. Indications for inclusion), you just mention the B-mode appearence of the prostate and the US-FNA but the clinical symptoms are completely forgotten. It is still a debate among specialist to consider if a dog having on heterogeneous prostate on ultrasound but no clinical signs has to be treated and can be considered as having a "clinical BPH". And because you don't mention the clinical symptoms, you don't correlate the clinical improvement with the modifications observed on ultrasound, which would have been very interesting. I understand that it was not your initial goal, but I find essential that you speak about the clinical aspects of the dogs enrolled in the study. It should be added and discussed.
Authors fully agree with the reviewer, the clinical data was collected and analysed simultaneously, but due to the huge amount of results it was decided to address and discuss clinical aspects in a separate paper. In the mean time when this manuscript was being prepared and submitted, the other paper concerning clinical aspects of the study has been published: https://doi.org/10.3390/ani10101936
The citation of this paper was added in this corrected version of the manuscript: Indication of inclusion Lines: 161-162 and Discussion Lines: 651-652. Authors hope it supports the data presented here and clarifies doubts regarding the clinical part of the study.
The word treatment was removed or replaced throughout manuscript. Also in the title we changed ‘treated’ to ‘receiving’ to address reviewer’s concerns.
Another frustrating point is that you didn't decide to add one group of BPH dogs that were surgically castrated. It should be also discussed.
Authors are grateful for this comment and completely agree with the reviewer. Authors plan to carry out studies which include castrated dogs as the ‘absolute control’. However, in this clinical trial the aim was to compare positive and negative control groups with two group of intact dogs receiving different pharmacological components. We mainly wanted to investigate the outcome depending on the pharmacological factor as the source of variation.
Now I am going to give a few statements line by line.
- lines 59-60: you state that to make a definitive diagnosis of prostate pathology a transabdominal prostatic biopsy is necessary. I don't fully agree. Of course, a prostatic biopsy appears as the gold standard, but FNA (which has quite a high correlation with biopsy) or even prostatic massage are often conclusive. Furthermore, I think there is a confusion in your mind between "FNA" and biopsy", as in "materials and methods" (line 164), you write that you were performing US-guided FNA, and in lines 167 and 169 you write "after prostate biopsy". I think this is contradictory and should be clarified;
The sentence in the Introduction was changed as follows: Although, for the definitive diagnosis of prostate pathology the transabdominal prostatic biopsy is the gold standard, the typical ultrasonographic picture together with the clinical findings, are usually enough for the presumptive diagnosis and in some cases suitable treatment [2].
The biopsy (lines: 167 and 169) was changed into FNA – authors are very sorry for such mistake and grateful for pointing it out.
- line lines 80-82: "it contains a steroidal androgen which... and inhibiting the action of 5a-reductase": please give a fundamental reference of this statement, as the exact mechanism of action of osaterone acetate is often unclear;
The reference was added
- line 106: table 1 (breeds) does not exist in what I received to review, as in what I had to review table 1 reports about the "subjective assessment of prostate gland parenchyma". Please add this table to your manuscript;
Authors are very sorry for overlooking this mistake. This article presents part of data collected for the trial. The clinical aspects were collected and analyzed simultaneously, but due to the abundance of data it was decided to present it in a separate paper. In the meantime, when this publication was edited and submitted, the other paper on clinical features has been published https://doi.org/10.3390/ani10101936 The citation of this paper was added in this corrected version of the manuscript: Indication of inclusion Lines: 161-162 and Discussion Lines: 651-652. Authors hope it supports the data presented here and clarifies doubts regarding the clinical part of the study.
- lines 109-111: group II (dogs with BPH but "no treatment administered", not even a placebo) raises an ethical concern to me. Either they were not having clinical signs - and therefore what you write on line 643 is wrong - or they were not clinically diseased, and it is the point I already spoke about. Anyway, if they had clinical signs, how did the owners accepted that their dogs were not treated ? Did they sign a written agreement ? What were the motivated explanations of your ethical committee to accept this ? And what about if these dogs had clinical signs appearing during the study or, if they had already clinical signs at the beginning, if these clinical signs had increased. Would they have been excluded from the study and fully treated ? You don't say anything about it. Please add more explanations.
The design of the study was consulted with pharmacologists, who recommended the positive control as a valuable group for comparison. As we regularly see many semen donors in our Clinic and Semen Bank, it seemed feasible. Especially, some owners, being afraid of the potential reduction in fertilizing capacity, sometimes initially refuse therapy or decide to postpone the BPH treatment.
lines 219-220: "whereas, in group IV ... 40% of the investigated dogs": this sentence is unclear, please could you rephrase it ? Espacially I don't understand what you mean when writing "it did not exceed more than 40% of the investigated dogs".
The sentence has been changed as follows: Whereas, in Group IV the echogenicity started to decrease as early as from D14 in 40% of the investigated dogs.
- figure 2 is interesting. It could be useful to add a similar figure (figure 2bis ?) with photos of the action of osaterone acetate.
The figure (Figure 3) was added.
- lines 494-495: you state that the mediastinum testis disappeared at W+24, but on the photo of the figure 5 (C), it is not evident.
Authors are sorry for this overstatement, the word ‘almost’ disappeared was added
- lines 500-505: a comment on the findings of testicularB-mode appearence of group IV during treatment is lacking.
The following sentence was added: Similarly, the imaging of the testicular parenchyma remained unchanged in the following observations.
- lines 552-554: "the RI indices were lower in group II ... whereas in group IV RI indices were lower in W8 of the study (p=0.02)": this is an interesting finding which should be discussed in "discussion", which potential explanations.
The sentence in Results section was rephrased: Only individual variances were noted in the RI indices, lower values were noted in Control Group II than in Group I on D14 (p=0,02) and in W+20 (p=0,01), whereas and in Group IV in W8 of the study (p=0,02), whereas in Group III RI remained similar throughout the study. The comments are presented in Discussion lines: 683-687 of the original manuscript.
- lines 598-599: already said: clinical data are lacking to claim a "treatment".
It was changed into: the study period
- line 643: already said: clinical symptoms are not mentioned in your manuscript.
Authors agree with the reviewer and changed the sentence as follows: Polisca et al. [14] investigated only German shepherd dogs of similar age and weight, whereas our investigations included dogs of various breeds, age and weight.
- lines 644- 645: I agree that - following you results - it seem interesting to combine OA and DA for the treatment of BPH but because no study has been done (it could have been another group of you study), I would prefer - line 644 - that you write: "it may be beneficial" rather than "it would be beneficial".
changed
- lines 704-705: I don't agree with the term "confirms". I would rather write "it tends to show" or a similar statement, not so affirmative.
changed
- line 720: already said: a group of surgically castrated dogs with BPH is lacking in your study and your choice not to include it should be discussed in "discussion".
The following sentence was added into Discussion (Lines: 758 -760): The reported US changes in the prostate noted with the use of pharmacological components should also be compared to the changes occurring in dogs, in which surgical castration was chosen as the BPH treatment, to see the difference between the presence or absence of gonadal steroidal feedback on the dynamics of changes in the prostate gland parenchyma.
Thank you very much.
Best regards.

Reviewer 2 Report
Thank you very much for the interesting study. Unfortunately, in my PDF File the figures are not all drawn correctly. This is essential to evaluate the figures…
Her are my comments:
Line 30: Abbreviations (SVP) should be not used or written out in the abstract
Line 52: Anatomical structures should be underlined by literature. The dog has two, the men has all four accessory glands.
Lines 113, 613: space between number and unit is missing (46.5kg -> 46.5_kg), please correct in the whole manuscript.
Lines 249-265: In this part of the table all values are equal. IS this correct? If yes, the table could be modified to make it easier to understand.
Lines 440, 569, 587: Here are problems with the figures as mentioned before
Author Response
Authors would like to thank the reviewer for comments and suggestions which helped to improve the MS. All suggested changes have been addressed as follows:
Line 30: Abbreviations (SVP) should be not used or written out in the abstract
Authors are sorry for this mistake, the full descriptions were added.
Line 52: Anatomical structures should be underlined by literature. The dog has two, the men has all four accessory glands.
Authors agree with the reviewer, we rephrased the sentence and added the reference to avoid any confusion: In dogs, the prostate is a bilobed gland located astride the prostatic urethra and surrounded by a capsule [1].
Lines 113, 613: space between number and unit is missing (46.5kg -> 46.5_kg), please correct in the whole manuscript.
corrected
Lines 249-265: In this part of the table all values are equal. IS this correct? If yes, the table could be modified to make it easier to understand.
Authors agree with the reviewer that the first part of the table presents the same information. Authors tried to modify it, according to the reviewer suggestions using SAA or ” sign. But in authors opinion then the table started to be unclear. We feel it is important to show in detail data presented in this table, so it was decided to return to the original version.
Lines 440, 569, 587: Here are problems with the figures as mentioned before
All figures has been replaced with new ones, as suggested by the reviewer.

Reviewer 3 Report
Review of the paper “B-Mode and Doppler ultrasonographic findings of prostate gland and testes in dogs treated with deslorelin or osaterone acetate”
This is an interesting paper which describes diagnostic imaging findings in 2 groups of dogs with prostatic hypertrophy treated with 2 different drugs, deslorelin and osaterone. I have provided a list of some key issues which should be addressed to improve the quality of the paper in the text below. I have also listed line where changes are advisble and/or the text should be removed or drastically altered
- The study design is clear and straightforward.
- Ultrasound images are valuable, of good quality and very interesting although not always completely clear; ultrasound figure legends are insufficient.
- A few citations are wrong or misplaced
- Some statements in the discussion section are inappropriate and should be deleted
- The level of scientific English must be improved. A few mistakes have been listed in the text below. However, although the text is understandable in most areas, the style is very often colloquial rather than scientific. I ahve attached the pdf file of the paper where I have – as an example - highlighted the sentences which need to be rephrased in the Introduction.
Line 49 – Change IMAGINING to Imaging
Lines 73-74 – Stiff gait is not a usual clinical signs. Although dogs with BPH may rarely show it, it is much more common in case of prostatitis rather than BPH.
Line 76 – hemospermia is not a cause for infertility. Dogs are widely known to be able to be fuly fertile when ejaculating bloody semen. The authors here cite the chapter Disorders of the Canine Prostate of the Canine and Feline Theriogenology on pages 337-355. However, in this chapter there is no information on a relationship between hemospermia and infertility, hemospermia is simply cited as a clinical sign of BPH. In a following chapter of the same book, Clinical approach to infertility in the male dog, on page 381 hemospermia is described as not necessarily causing infertility.
Line 76-78 – the words “apart from surgical castration” should be removed. Castration as a treatment for dog with BPH has been a source of debate over the last 20 years since the paper from Teske et al (Molecular and Cellular Endocrinology 197 (2002) 251-/255) was published. Even though there is still not enough evidence that castration in older male dogs plays a role in the development of prostatic adenocarcinoma, the above paper has created a lot of discussion and the issue has become controversial. As such, simply stating nowadays that dogs with BPH can be castrated is an oversimplification of a very complicated issue. The whole sentence at lines 76-78 should be rephrased.
Lines 82 – The authors should provide a reference for the statement that osaterone acetate inhibits the action of 5a-reductase
Line 154 – I believe the word “imagined” is not appropriate in this context and should be changed to “imaged”
Line 184 – the sentence “to see treatment onset and duration” is unclear and should be rephrased
Line 211 – “did not changed” should be “did not change”
Line 214 – the highlighted sentence should be rephrased either by removing WITH or by placing the word MARGINS after “with”
Line 221 – change investigate to investigated
Line 227 – the ultrasound picture of Letter E in the figure legend is not clear, the distance between the 2 symbols seems very long to be defined a ”small” cyst. Also, it is not clear where is the prostatic lobe with respect to this anechoic cyst. A scale in cm should be provided, better if in all pictures
Lines 230-236 – It is assumed that ultrasound images of letters A and B correspond to a single dog each, however that is not clear and should be specified. The prostates in A1 and A2 seem to have a higher echogenicity while the image A3 has a more normal echogenicity but A4 is different, the prostatic margins are not clearly identifiable and the reader is left with a question mark. A similar lack of clarity can be seen in B images, whereby B1 is clear but prostatic margins in B2, B3 and B4 are not identifiable. Besides clarifying these doubts the authors should also explain what is indicated in between the symbols.
Lines 437-444 – Why are some of the red dots outside of the graphs?
Lines 496-499 – The sentence is awkward and should be rephrased; the word “changed” should be “change”
Lines 506-511 – The text in the legend refers to A, B and C but instead it should use the numbers 1, 2 and 3 as in the related ultrasound images.
Lines 587-591 – The graphs are not aligned
Lines 597-599 – This sentence is a bit naive and obvious as ultrasound has been used for decades for the diagnosis and follow-up of canine BPH
Line 631 – Osaterone is commonly known as a progesterone derivative. The authors should provide a reference for this statement
Line 634 – Change echogenity to echogenicity
Line 634 and followings – The discussion here is valuable but readers get confused as in the 2 sentence (“The gland echogenity….” and “Similar results….”) one might think that the authors are discussing about the same treatment (osaterone acetate) and instead reference 22 by Goericke-Pesch concerns the use of of a GnRH agonist (nafarelin, not even deslorelin), similary to papers n. 24 and 14 (these on deslorelin) cited in the next 2 sentences.
Lines 644-646 – this statement leads readers astray as it sound as if the 2 treatments OA+DA were used together by the authors. The sentence should be rephrased suggesting that this option deserves future studies
686-687 – Starting a sentence with “Opposite to dog,” is a rather colloquial way of expressing thoughts; this is not immediately clear to readers. The sentence should be rephrased to something lie “This is contrary to what happens in dogs, as…:”
Lines 705 – Replace changed with change
Line 721 – Change chose to choose
Lines 721-724 – The sentence “Indications for the use of…..” is awkward and should be rephrased
Lines 724-734 – A totally new protocol cannot be proposed at the end of a research paper which describes two different protocols. The word “guideline” is used inappropriately here and should be removed as guidelines are normally based on evidence. The suggestion of the authors of this paper for a new clinical management of BPH makes sense and could be agreed upon, but cannot be proposed in this context. This entire section must be deleted.

Author Response
Authors would like to thank the reviewer for comments and suggestions which help to improve the MS. All suggested changes have been addressed as follows:
This is an interesting paper which describes diagnostic imaging findings in 2 groups of dogs with prostatic hypertrophy treated with 2 different drugs, deslorelin and osaterone. I have provided a list of some key issues which should be addressed to improve the quality of the paper in the text below. I have also listed line where changes are advisable and/or the text should be removed or drastically altered
- The study design is clear and straightforward.
- Ultrasound images are valuable, of good quality and very interesting although not always completely clear; ultrasound figure legends are insufficient.
- A few citations are wrong or misplaced
- Some statements in the discussion section are inappropriate and should be deleted
- The level of scientific English must be improved. A few mistakes have been listed in the text below. However, although the text is understandable in most areas, the style is very often colloquial rather than scientific. I ahve attached the pdf file of the paper where I have – as an example - highlighted the sentences which need to be rephrased in the Introduction.
The text was corrected and proof read according to the reviewer suggestions.
Line 49 – Change IMAGINING to Imaging
Authors are very sorry for this mistake, it was corrected.
Lines 73-74 – Stiff gait is not a usual clinical signs. Although dogs with BPH may rarely show it, it is much more common in case of prostatitis rather than BPH.
Authors agree with the reviewer and removed the phrase
Line 76 – hemospermia is not a cause for infertility. Dogs are widely known to be able to be fuly fertile when ejaculating bloody semen. The authors here cite the chapter Disorders of the Canine Prostate of the Canine and Feline Theriogenology on pages 337-355. However, in this chapter there is no information on a relationship between hemospermia and infertility, hemospermia is simply cited as a clinical sign of BPH. In a following chapter of the same book, Clinical approach to infertility in the male dog, on page 381 hemospermia is described as not necessarily causing infertility.
Authors are very sorry for this mistake, the sentence was removed.
Line 76-78 – the words “apart from surgical castration” should be removed. Castration as a treatment for dog with BPH has been a source of debate over the last 20 years since the paper from Teske et al (Molecular and Cellular Endocrinology 197 (2002) 251-/255) was published. Even though there is still not enough evidence that castration in older male dogs plays a role in the development of prostatic adenocarcinoma, the above paper has created a lot of discussion and the issue has become controversial. As such, simply stating nowadays that dogs with BPH can be castrated is an oversimplification of a very complicated issue. The whole sentence at lines 76-78 should be rephrased.
Removed according to the reviewer suggestions, now the sentence is as follows: In dogs with BPH, pharmacological therapy should be considered, especially in valuable studs, where preserving fertility is crucial.
Lines 82 – The authors should provide a reference for the statement that osaterone acetate inhibits the action of 5a-reductase
The reference was added
Line 154 – I believe the word “imagined” is not appropriate in this context and should be changed to “imaged”
Corrected, authors are very sorry for this mistake
Line 184 – the sentence “to see treatment onset and duration” is unclear and should be rephrased
It was removed and the sentence is now as follows: the results were also compared between Group III and IV to see differences between both investigated drugs.
Line 211 – “did not changed” should be “did not change”
corrected
Line 214 – the highlighted sentence should be rephrased either by removing WITH or by placing the word MARGINS after “with”
Corrected as follows: with margins well differentiated from the surrounding tissues
Line 221 – change investigate to investigated
corrected
Line 227 – the ultrasound picture of Letter E in the figure legend is not clear, the distance between the 2 symbols seems very long to be defined a ”small” cyst. Also, it is not clear where is the prostatic lobe with respect to this anechoic cyst. A scale in cm should be provided, better if in all pictures
The picture was replaced, authors thank for this suggestion, the original picture was misleading. Unfortunately, we are not able to provide the scale as it was not saved with the original image.
Lines 230-236 – It is assumed that ultrasound images of letters A and B correspond to a single dog each, however that is not clear and should be specified. The prostates in A1 and A2 seem to have a higher echogenicity while the image A3 has a more normal echogenicity but A4 is different, the prostatic margins are not clearly identifiable and the reader is left with a question mark. A similar lack of clarity can be seen in B images, whereby B1 is clear but prostatic margins in B2, B3 and B4 are not identifiable. Besides clarifying these doubts the authors should also explain what is indicated in between the symbols.
The Figure’s description was improved as follows:
Figure 2. Changes in B-mode prostate imaging in dogs receiving deslorelin acetate, visible gradual involution of multiple cysts and changes in echogenicity of prostatic parenchyma. Letters A and B correspond to a single patient in the following examinations on days: 1 – day 0; 2 – week 8; 3 – week 16, 4 – week 24.
A – dog with large and multiple parenchymal cysts (> 10 mm), sagittal view, right lobe indicated in pictures;
B – dog with small and multiple parenchymal cysts (< 10 mm), transverse view, the prostate gland indicated in pictures;
Lines 437-444 – Why are some of the red dots outside of the graphs?
In original version all graphs were ok, the dots were shifted during the submitting process. All graphs were replaced, to avoid such problems in the future.
Lines 496-499 – The sentence is awkward and should be rephrased; the word “changed” should be “change”
It was changed into: In Group IV the focal changes and focal fibrosis noted on D0 in some dogs did not change in the following US examinations throughout the whole observation period.
Lines 506-511 – The text in the legend refers to A, B and C but instead it should use the numbers 1, 2 and 3 as in the related ultrasound images.
Corrected
Lines 587-591 – The graphs are not aligned
Corrected
Lines 597-599 – This sentence is a bit naive and obvious as ultrasound has been used for decades for the diagnosis and follow-up of canine BPH
The sentence was changed into: The presented results confirm the clinical efficacy of ultrasound examination for diagnosis and treatment monitoring of BPH in dogs.
Line 631 – Osaterone is commonly known as a progesterone derivative. The authors should provide a reference for this statement
The statement was changed into: the osaterone acetate, is a steroid chemically related to progesterone,
Line 634 – Change echogenity to echogenicity
Changed
Line 634 and followings – The discussion here is valuable but readers get confused as in the 2 sentence (“The gland echogenity….” and “Similar results….”) one might think that the authors are discussing about the same treatment (osaterone acetate) and instead reference 22 by Goericke-Pesch concerns the use of of a GnRH agonist (nafarelin, not even deslorelin), similary to papers n. 24 and 14 (these on deslorelin) cited in the next 2 sentences.
The names of the medications used by each author were added in the text.
Lines 644-646 – this statement leads readers astray as it sound as if the 2 treatments OA+DA were used together by the authors. The sentence should be rephrased suggesting that this option deserves future studies
Changed into: The obtained results suggest that in cases with severe BPH symptoms it may be beneficial to initiate the treatment using both OA and DA together, to achieve a quick clinical response (OA) which last longer (DA), comparing to the individual use of these drugs, but this needs further research.
686-687 – Starting a sentence with “Opposite to dog,” is a rather colloquial way of expressing thoughts; this is not immediately clear to readers. The sentence should be rephrased to something lie “This is contrary to what happens in dogs, as…:”
Corrected according to reviewer suggestions
Lines 705 – Replace changed with change
corrected
Line 721 – Change chose to choose
corrected
Lines 721-724 – The sentence “Indications for the use of…..” is awkward and should be rephrased
removed
Lines 724-734 – A totally new protocol cannot be proposed at the end of a research paper which describes two different protocols. The word “guideline” is used inappropriately here and should be removed as guidelines are normally based on evidence. The suggestion of the authors of this paper for a new clinical management of BPH makes sense and could be agreed upon, but cannot be proposed in this context. This entire section must be deleted.
removed
